# Determinants of the Consumption of Regular Soda, Sport, and Energy Beverages in Spanish Adolescents

**DOI:** 10.3390/nu13061858

**Published:** 2021-05-29

**Authors:** Helmut Schröder, Vanessa Cruz Muñoz, Marta Urquizu Rovira, Victoria Valls Ibañez, Josep-Maria Manresa Domínguez, Gerard Ruiz Blanco, Monserrat Urquizu Rovira, Pere Toran Monserrat

**Affiliations:** 1Cardiovasclar Risk and Nutrition Research Group, Institut Hospital del Mar d’Investigacions Mèdiques, 08003 Barcelona, Spain; HSchoeder@imim.es; 2CIBER Epidemiology and Public Health, 28029 Madrid, Spain; 3Metropolitana Nord Research Support Unit, IDIAP-JGol, 08290 Cerdanyola del Vallès, Spain; vcruzm.mn.ics@gencat.cat (V.C.M.); murquizu.ics@gencat.cat (M.U.R.); vvalls.mn.ics@gencat.cat (V.V.I.); ptoran.bnm.ics@gencat.cat (P.T.M.); 4Sabadell Nord Primary Care Team, Institut Català de la Salut, 08203 Sabadell, Spain; 5Serraperera Primary Care Team, Institut Català de la Salut, 08290 Cerdanyola del Vallès, Spain; 6La Serra Primary Care Team, Institut Català de la Salut, 08202 Sabadell, Spain; 7Department of Nursing, Universitat Autònoma de Barcelona, Cerdanyola del Vallès 08193, Spain; 8Multidisciplinary Research Group in Health and Society (GREMSAS), 08010 Barcelona, Spain; 9Rambla Primary Care Center, Mútua de Terrassa, 08221 Terrassa, Spain; gerard.ru94@gmail.com; 10Emergency Care, Hospital Vall d’Hebrón, Institut Català de la Salut, 08035 Barcelona, Spain; murquizu@vhebron.net

**Keywords:** adolescents, sweetened beverages, healthy behaviour, energy drinks, carbonated drinks, cross-sectional studies

## Abstract

Increasing sugar-sweetened beverages (SSB) consumption and associated health impacts warrant health-policy action. We assessed associations of socioeconomic and lifestyle variables with adolescents’ consumption of regular soda (RSD), sport (SD), and energy (ED) drinks. Cross-sectional study of 3930 Spanish adolescents (2089 girls, 1841 boys) aged 13–18 years). We compared frequency of consuming each SSB type (European Food Safety Authority questionnaire) with sociodemographic and lifestyle variables (standardized questions). RSD, SD, and ED were consumed at least weekly by 72.7%, 32.3%, and 12.3% of participants, respectively, and more frequently (*p* < 0.001) by boys, compared to girls. Multivariate ordinal logistic regression showed inverse association between RSD, SD, and ED consumption and parental occupation-based socioeconomic status (*p* < 0.01). Daily smoking was associated (*p* < 0.001) with higher ED (OR 3.64, 95% CI 2.39–5.55) and RSD (OR 2.15, 95% CI 1.56–2.97) consumptions. SD intake was associated inversely with smoking (OR 0.60, 95% CI 0.40–0.89, *p* = 0.012) and directly with physical activity (OR 2.93, 95% CI 2.18–3.95, *p* < 0.001). School performance was lower among ED (OR 2.14, 95% CI, 1.37–3.35, *p* = 0.001) and RSD (OR 1.81, 95% CI 1.24–2.64, *p* = 0.002) consumers, compared to SD. Maleness and low socioeconomic status predicted SSB consumption. Smoking and low school performance were associated with higher ED and RSD intakes.

## 1. Introduction

The consumption of sugar-sweetened beverages (SSB) is widespread in the child population [1]. This is of concern due to the associated adverse health outcomes observed in children and adolescents [2,3,4]. The high sugar content and absence of other essential nutrients in these beverages can substantially increase energy intake without contributing any additional significant dietary value. Therefore, the consumption of these beverages worsens diet quality and is suspected to be one of the main drivers of the childhood obesity epidemic [4,5].

Sport drinks (SD), energy drinks (ED), and regular soda drinks (RSD) are subtypes of SSB and currently under discussion due to their potential health risks [6]. The consumption of sport drinks is recommended to meet the increased need of glucose during prolonged high intensity exercise [7]. However, this makes little sense for individuals engaging in less strenuous physical activity or in children at school. Furthermore, as the recommended sodium intake is met with a healthy diet [7], the additional supply of sodium by these drinks might lead to an excessive intake of this nutrient.

Energy drink consumption by children and adolescents has received attention due to its increasing popularity and the potential caffeine toxicity [8,9] associated with the consumption of these beverages. In Spain, about 60% of adolescents and 26% of children reported ED consumption [10]. This is especially of concern due to increasing evidence that the consumption of these beverages increases cardiovascular risk not only in adults but also in youth [7,11].

In order to develop successful childhood intervention programs aimed to promote healthy dietary habits, it is important to identify children who are particularly vulnerable to consuming SSB. The hypothesis of the present work was that SSB consumption is related to unfavorable lifestyle habits and low maternal socioeconomic status. The present study provides data on sociodemographic and lifestyle determinants associated with SSB consumption, including RSD, ED, and SD, in a large population of Spanish adolescents aged 13 to 18 years. 

## 2. Methods

### 2.1. Design and Study Population

This cross-sectional study is part of the “consumption of regular soda, isotonic, and energy beverages in Spanish adolescents” (BEENIS) project. The project objective was to determine the prevalence, amount, and habits associated with SSB consumption among the school population of Sabadell, a city within the Metropolitan Area of Barcelona. The population was divided into 3 age groups (6–9 years, 10,148 students; 10–12 years, 7035 students; and 13–18 years, 9146 students) with a specific questionnaire for each age group (Figure 1). All schools were contacted and invited to participate. The project was approved by the IDIAP Jordi Gol Research Ethics Committee (code P15/113) and the consent of all parents and of students over 13 years of age was obtained. For the present study, we selected 3930 of 9146 invited participants (42.9%) for analysis (Figure 1). Inclusion criteria were adolescents aged 13 to 18 years with complete data on SSB consumption. 

### 2.2. Data Collection

The BEENIS questionnaire was self-administered in the classrooms under teacher supervision. Students were asked how often they consumed RSD, ED, or SD during the last year. The questions related to SSB consumption were taken from the European Food Safety Authority (EFSA) questionnaire administered to gather data on consumption of energy drinks by specific consumer groups [10]. The six response options were “never”, “1 or 2 times/month”, “once a week”, “2–3 days a week”, “4–5 days a week” and “every day”. For the purpose of analysis, we reduced the responses to four categories: “never”, “less than 2 times a month”, “occasionally [1–5 times/week]”, and “usually [>5 times/week]. An identical structure was used for specific questions on the consumption of each type of SSB (SD, ED, and RSD). Participants also responded to questions on frequency of sport activities, smoking (never, occasionally, regular), sleep duration of at least 8 h/day (yes/no), and screen viewing including television, computer, play station, and tablet (never, occasionally, less than 1 h per day, at least 1 h but less than 3 h per day, between 3 h and less than 5 h per day, and more than 5 h per day). They were also asked how many subjects they had failed in the previous school year and their maternal country of origin and parental occupation. Low school performance was defined as failing more than 3 subjects in the previous school year [12]. Categories of occupation-related socioeconomic status followed the criteria of the National Health Survey of the Ministry of Health, Consumption and Social Welfare of Spain [13]. The survey establishes 6 social categories, which were reduced for our analysis to 3 socioeconomic status groups: high (categories 1 + 2: managers, senior technicians, artists and athletes), medium (categories 3 + 4: middle managers and skilled workers) and low (categories 5 + 6: unskilled workers and others).

A pilot study including 280 children from a school in a nearby town was performed to identify potential problems in understanding the questions included in the questionnaire. 

### 2.3. Statistical Analysis

The qualitative variables are summarized with their absolute and relative frequency, and the continuous variables with their mean and standard deviation. Differences between categorical and continuous variables were determined by Pearson Chi Square and Student t test for independent data, respectively.

Ordered logistic regression models were fitted to determine the association between SSB consumption and lifestyle and sociodemographic variables. In the first bivariate regression models, we identified variables that were individually associated with the three types of SSB consumption (SD, ED, RSD) at *p* < 0.10. These variables were then included in multivariate models. The final models were mutually adjusted for the type of SSB consumption; the Akaike information criterion and biological plausibility were also taken into account. Statistical significance was set at *p* < 0.05 (two-tailed). Data were analysed with the Statistical Package Stata MP 15.1 (StataCorp LLC, Texas, USA) for Windows.

## 3. Results

Girls were somewhat younger, less physically active, and less likely to smoke and to have low school performance than boys (Table 1). Both the frequency and amount of SD, ED, and RSD were higher in boys, compared to girls. 

Multivariate ordinal logistic regression revealed that the frequency of the consumption of these beverages increased with male gender, smoking frequency, low school performance, maternal non-Spanish origin, and low parental occupation-based socioeconomic classification (Table 2). High screen time and high levels of physical activity were associated with an increasing consumption of RSD and SD, respectively (Table 2). These associations remained stable and significant in multivariate ordinal regression models with and without mutual adjustment for other types of SSB consumption (Table 3). The frequency of the consumption of any type of SSB increased by concomitant consumption of the other two types (Table 3). 

## 4. Discussion

The main finding of this analysis was that the frequency of consumption of SSB was positively associated with male gender and low occupation-based socioeconomic status. The consumption of RSD and ED was directly related to smoking, low school performance, and low levels of physical activity. In contrast, adolescents who consumed SD frequently were more physically active and less likely to smoke compared to those with less frequent SD consumption. 

Evidence indicates that the consumption of SSB is associated with several adverse health outcomes in adults and youth [2,3,7,8,9]. Therefore, it is not surprising that the European Academy of Paediatrics and the European Childhood Obesity Group recently proposed several strategies with the aim to reduce SSB consumption in youth [1].

SSB encompasses a wide range of beverage types [14]. Whereas there is considerable evidence on determinants of SSB consumption in youth [15], less is known for individual types of SSB. The present study focused on three types of SSB consumption, namely regular sodas, sport drinks, and energy drinks. Energy drinks are of particular interest due to their high caffeine content and increasing popularity among adolescents [16]. The caffeine content of these beverages is four times as high as a regular cola drink and similar to that of coffee [7]. This is of concern due to the clinical effects of caffeine such as an increase in heart rate and blood pressure, among others [17,18]. In the present study, male gender strongly determined the consumption of SSB, having the strongest impact on SD consumption, followed by ED and RSD. Mutual adjustment of these beverages in addition to other sociodemographic and lifestyle variables did not affect this finding, indicating an independent impact of gender on SSB consumption. This finding is in line with previous research [11,19,20]. Both the Norwegian study on a cohort of adolescents aged 15 to 17 years [19] and Spanish study in a large sample of Spanish adolescents aged 14 to 18 years [20] further identified male gender as the most important variable associated with ED consumption. In the present study, the consumption of SSB was strongly associated with smoking, but in different directions: SD consumers were more likely to be non-smokers whereas the opposite was found for ED and RSD. Furthermore, it is of interest to note that school performance was not affected by SD consumption but was strongly compromised by RSD and ED consumption. Finally, physical activity was positively associated with SD consumption. These findings indicate that the consumption of sport drinks was embedded in a somewhat healthier lifestyle, compared to ED and RSD in the present population.

Evidence, albeit scarce, indicates that parental socioeconomic status -whether classified by educational or occupational level- is a determinant for SSB consumption by children and adolescents [19,20,21,22,23]. Recent findings in Norwegian adolescents show that those with mothers having more education were less likely to consume SSB, compared to those with mothers having less education [19]. Similarly, a Spanish cohort study revealed a strong inverse association between maternal educational level and consumption of SSB [20]. In the present analysis, low parental occupational level was associated with higher consumption of all three SSB types studied. Socioeconomic status is an important determinant of an unhealthy lifestyle, including SSB consumption, in adults [24,25]. This, in addition to the pivotal role of parental habits on a child’s lifestyle, might partially explain the relationship between parental socioeconomic status and a child’s SSB consumption. Furthermore, De Coen and colleagues found that parental practices (e.g., having SSB in the home, SSB served at meals, allowing children access to SSB whenever they want it) mediates the association between maternal educational level and child SSB consumption [21]. It is notable that school performance decreased with SSB consumption in the present population, which is in line with previous research [19,26] Whether this relationship is mediated by the nutrient composition of SSB or parental socioeconomic variables or both is unclear [26].

The results of the present study underline the need for the implementation of health policies with the aim to reduce SSB consumption. Taxation of SSB consumption might be a tool for the reduction of the consumption of these beverages. Currently SSB taxes are implemented in more than 40 countries [27]. To date it is too early for a final conclusion of the efficacy of this measure, however, current data are promising [28,29].

The strengths of the present study include the relatively large sample size and the analysis of SSB consumption stratified by SD, ED, and RSD. In addition, final mutual adjustment of these beverage types provided insights into the independent relationship between these beverages and sociodemographic and lifestyle variables. A limitation of the present study is the use of a non-validated questionnaire of SSB consumption. Furthermore, due to limited time data of physical activity, school performance, and screen time were measured by short not validated questions. This fact could have increased the measurement error inherent in self-reported data.

## 5. Conclusions

Results of the present study showed that SSB consumption was more pronounced in adolescent boys than girls and was more embedded in families with low socioeconomic status than in their peers with higher socioeconomic status. Furthermore, the frequency of ED and RSD consumption increased with smoking and lower physical activity levels. The opposite was observed for SD. Due to the adverse effect of SSB on child cardiometabolic health, policies that aim to reduce the consumption of these beverages is warranted. Indeed, higher SSB taxes seem to significantly reduce their consumption [27,28].

## Figures and Tables

**Figure 1 nutrients-13-01858-f001:**
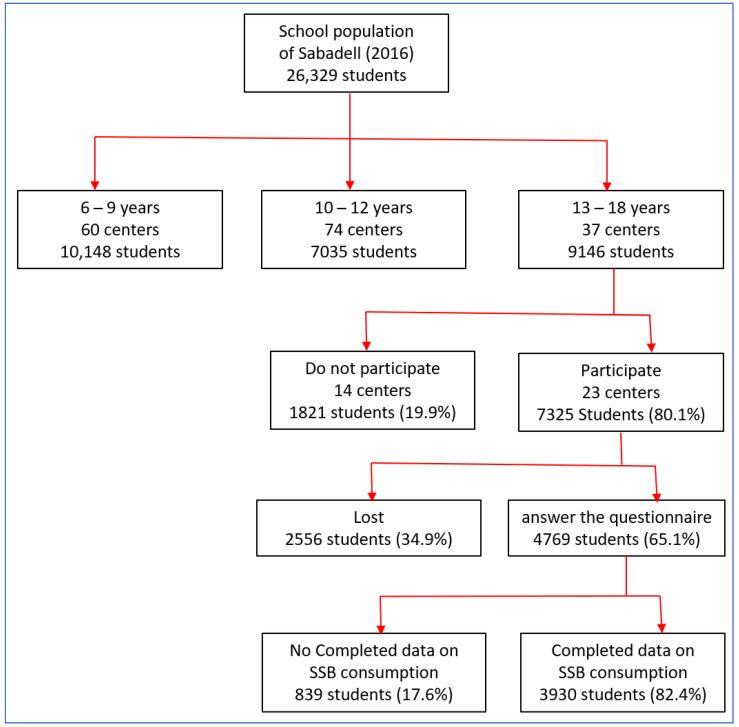
Participation flow chart from Beenis Study. This diagram shows the flow of participants in this article. Initially, all schools in Sabadell were invited to participate (*n* = 26,329 students). In the group of 9146 students between 13–18 years, 23 centers with a population of 7325 students agreed to participate, of which 4769 answered the questionnaire and 3930 had Completed data on SSB consumption. SSB: sugar-sweetened beverages.

**Table 1 nutrients-13-01858-t001:** Characteristics of participants.

	Boys	Girls	All	*p*	Missing (%) ^1^
	1841 (46.8%)	2089 (53.2%)	3930		403 (8.50%
Age (years)	14.8 (1.4)	14.8 (1.5)	14.8 (1.5)	0.086	91 (1.91%)
Maternal origin				0.145	105 (2.20%)
- Spain	1510 (82.0%)	1745 (83.5%)	3255 (82.8%)		
- Other	331 (18%)	344 (16.5%)	675 (17.2%)		
Paternal occupation ^1^				0.860	817 (17.10%)
- High	348 (22.3%)	393 (22.1%)	741 (22.2%)		
- Medium	578 (37.0%)	674 (37.9%)	1252 (37.5%)		
- Low	636 (40.7%)	711 (40.0%)	1347 (40.3%)		
Maternal occupation ^2^				0.344	943 (19.80%)
- High	369 (24.4%)	391 (22.4%)	760 (23.4%)		
- Medium	537 (35.6%)	622 (35.7%)	1159 (35.6%)		
- Low	604 (40.0%)	731 (41.9%)	1335 (41.0%)		
Sleep duration (>8 h/day)	794 (43.1%)	1101 (52.7%)	1895 (48.2%)	<0.001	130 (2.70%)
Smoking (%)					56 (1.20%)
- Nonsmoker	1573 (85.4%)	1689 (80.9%)	3262 (83%)	<0.001	
- Occasional	268 (10.4)	256 (12.2%)	447 (11.4%)	<0.001	
- Daily	77 (4.2%)	144 (6.9%)	221 (5.6%%)		
Screen viewing (>2 h/day)	1620 (88%)	1860 (89%)	3480 (88.5%)	0.306	119 (2.50%)
Physical activity ^3^	1644 (89.3%)	1646 (78.8%)	3290 (83.7%)	<0.001	99 (2.10%)
Low school performance ^4^	128 (7.0%)	75 (3.6%)	203 (5.2%)	<0.001	120 (2.50%)
Consumption of SSB					
- Regular soda					31 (0.70%)
Less than one/week	373 (20.3%)	696 (33.4%)	1069 (27.3%)	<0.001	
One to five/week	1126 (61.3%)	1115 (53.5%)	2241 (57.2%)		
Daily	337 (18.4%)	274 (13.1%)	611 (15.6%)		
- Sport drinks					42 (0.90%)
Less than one/week	977 (53.4%)	1674 (80.4%)	2651 (67.7%)	<0.001	
One to five/week	767 (41.9%)	380 (18.2%)	1147 (29.3%)		
Daily	87 (4.8%)	29 (1.4%)	116 (3%)		
- Energy drinks					38 (0.80%)
Less than one/week	1514 (82.6%)	1916 (92.1%)	3430 (87.7%)	<0.001	
One to five/week	287 (15.7%)	144 (6.9%)	431 (11.0%)		
Daily	31 (1.7%)	21 (1.0%)	52 (1.3%)		

SSB: sugar-sweetened beverages. Variables are expressed in absolute numbers and proportions. ^1^ Missing records in each variable (*n* = 4769) ^2^ Occupation-based socio-economic classification, based on the categories in National Health Survey of the Ministry of Health, Consumption and Social Welfare of Spain (13): high = categories 1 + 2 (managers, senior technicians, artists and athletes), medium = categories 3 + 4 (middle managers and skilled workers) and low = categories 5 + 6 (unskilled workers and others). ^3^ Physically active every day. ^4^ Failed more than 3 subjects in the previous school year.

**Table 2 nutrients-13-01858-t002:** Association between sugar-sweetened beverage (SSB) consumption and sociodemographic and lifestyle variables. Multivariate ordinal logistic regression with SSB consumption as the dependent variable. Exposure variables were mutually adjusted.

	Regular Soda		Sport Drinks		Energy Drinks	
	OR (95% CI)	*p*	OR (95% CI)	*p*	OR (95% CI)	*p*
Gender						
- Girls	Reference		Reference		Reference	
- Boys	1.91 (1.64–2.22)	<0.001	3.29 (2.77–3.90)	<0.001	2.64 (2.05–3.40)	<0.001
Maternal origin						
- Spain	Reference		Reference		Reference	
- Other	1.34 (1.08–1.66)	0.008	1.26 (0.99–1.60)	0.055	1.75 (1.30–2.37)	<0.001
Screen time ^1^	1.85 (1.47–2.32)	<0.001	1.06 (0.81–1.37)	0.679	1.21 (0.81–1.79)	0.355
Smoking						
- Nonsmoker	Reference		Reference		Reference	
- Occasional	1.56 (1.23–1.96)	<0.001	1.20 (0.92–1.56)	0.177	2.18 (1.56–3.03)	<0.001
- Daily	2.60 (1.89–3.57)	<0.001	0.84 (0.57–1.25)	0.387	3.65 (2.42–5.52)	<0.001
School performance ^2^	1.99 (1.37–2.89)	<0.001	1.56 (1.04–2.33)	0.031	2.12 (1.36–3.29)	0.001
Physical activity ^3^	0.94 (0.76–1.15)	0.544	2.89 (2.15–3.87)	<0.001	1.25 (0.88–1.77)	0.223
Paternal occupation ^4^						
- High	Reference		Reference		Reference	
- Medium	1.22 (1.00–1.48)	0.051	1.34 (1.06–1.69)	0.015	1.40 (0.96–2.04)	0.079
- Low	1.71 (1.40–2.10)	<0.001	1.52 (1.20–1.93)	0.001	1.78 (1.23–2.59)	0.002
Maternal occupation ^4^						
- High	Reference		Reference		Reference	
- Medium	1.30 (1.07–1.58)	0.008	1.27 (1.01–1.59)	0.041	1.23 (0.86–1.78)	0.259
- Low	1.56 (1.27–1.91)	<0.001	1.40 (1.10–1.77)	0.005	1.72 (1.21–2.45)	0.003

^1^ More than 2 h every day. ^2^ Failed more than 3 subjects in the previous school year. ^3^ Physically active every day. ^4^ Occupation-based socio-economic classification, based on the categories in National Health Survey of the Ministry of Health, Consumption and Social Welfare of Spain (13): high = categories 1 + 2 (managers, senior technicians, artists and athletes), medium = categories 3 + 4 (middle managers and skilled workers) and low = categories 5 + 6 (unskilled workers and others).

**Table 3 nutrients-13-01858-t003:** Association between mutually adjusted sugar-sweetened beverage consumption and sociodemographic and lifestyle variables.

	Regular Soda		Sport Drinks		Energy Drinks	
	OR (CI 95%)	*p*	OR (CI 95%)	*p*	OR (CI 95%)	*p*
Gender						
- Girls	Reference		Reference		Reference	
- Boys	1.51 (1.29–1.76)	<0.001	2.95 (2.48–3.51)	<0.001	2.10 (1.62–2.72)	<0.001
Maternal origin						
- Spain	Reference		Reference		Reference	
- Other	1.17 (0.94–1.46)	0.152	1.09 (0.85–1.39)	0.508	1.66 (1.22–2.25)	0.001
Screen time ^1^	1.66 (1.32–2.10)	<0.001	0.93 (0.71–1.22)	0.614	1.11 (0.74–1.65)	0.623
Smoking						
- Nonsmoker	Reference		Reference		Reference	
- Occasional	1.29 (1.02–1.64)	0.037	0.90 (0.69–1.18)	0.447	2.15 (1.53–3.00)	<0.001
- Daily	2.15 (1.56–2.97)	<0.001	0.60 (0.40–0.89)	0.012	3.64 (2.39–5.55)	<0.001
School performance ^2^	1.81 (1.24–2.64)	0.002	1.35 (0.90–2.03)	0.148	2.14 (1.37–3.35)	0.001
Physical activity ^3^	0.83 (0.68–1.03)	0.094	2.93 (2.18–3.95)	<0.001	1.07 (0.74–1.53)	0.733
Paternal occupation ^4^						
- High	Reference		Reference		Reference	
- Medium	1.17 (0.96–1.43)	0.123	1.28 (1.01–1.63)	0.043	1.42 (0.97–2.08)	0.070
- Low	1.62 (1.32–1.99)	<0.001	1.42 (1.11–1.81)	0.005	1.69 (1.16–2.46)	0.006
Maternal occupation ^4^						
- High	Reference		Reference		Reference	
- Medium	1.27 (1.04–1.54)	0.018	1.26 (1.00–1.59)	0.055	1.23 (0.85–1.77)	0.277
- Low	1.40 (1.14–1.72)	0.001	1.27 (1.00–1.61)	0.054	1.68 (1.18–2.41)	0.004
Regular soda drinks	-------		3.77 (2.17–6.55)	<0.001	2.92 (1.06–8.07)	0.039
Sport drinks	1.89 (1.60–2.23)	<0.001	-----------		3.00 (2.17–4.14)	<0.001
Energy drinks	2.02 (1.71–2.39)	<0.001	2.50 (2.09–2.99)	<0.001	-----------	

Multivariate ordinal logistic regression with SSB consumption as the dependent variable. Exposure and dependent variables were mutually adjusted. ^1^ More than 2 h every day. ^2^ Failed more than 3 subjects in the previous school year. ^3^ Physically active every day. ^4^ Occupation-based socio-economic classification, based on the categories in National Health Survey of the Ministry of Health, Consumption and Social Welfare of Spain (13): high = categories 1 + 2 (managers, senior technicians, artists and athletes), medium = categories 3 + 4 (middle managers and skilled workers) and low = categories 5 + 6 (unskilled workers and others).

## Data Availability

The data supporting the conclusions of this study are available upon reasonable request and under the supervision of IDIAP Jordi Gol.

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
