# Peer review of "Determinants of the Consumption of Regular Soda, Sport, and Energy Beverages in Spanish Adolescents"

_nutrients, 2021, doi:10.3390/nu13061858_

Round 1

Reviewer 1 Report

The manuscript reported analysis results with a cross-sectional data collected among 13-18 years old European adolescents living in Spain. The frequencies of children’s self-reported consumptions of regular soda, sport and energy drinks were compared with sociodemographic and lifestyle factors. Results indicated male and low socioeconomic stratus predicted the sugar-sweeten beverage consumption, and smoking and low school performance were associated with higher energy drink and regular soda intakes. The research findings have certain degree of importance of public health implications. The analysis is technically sound. The reviewer has the following suggestions for the authors’ consideration in order to improve the quality of manuscript.

  • On page 2 line 77, data from 3930 of 9146 invited participants (42.9%) were used for the analysis. The inclusion criteria were adolescents aged 13 to 18 years with complete data on SSB consumption. It would be better to provide detailed sample size information for each age groups (i.e. 6-9 years, 10-12 years and 13-18 years) and percentage of missing data for the variables using in the analysis.
  • How was the sampling strategy used in this study? Should the potential intra-class correlations of behavioral outcomes due to students nested within school be considered in the analysis?
  • Rough measurements of physical activity, school performance and screen time were used without justification of measurement methodology and validity. What is the rationale to use failing 3 subjects in the previous school year as an indicator for low school performance? Other than TV watching, video gaming and computer/tablet use should also be considered for screen time estimate for physical inactivity.  
  • What was the theory or theoretical framework to guide this study and to interpret the analysis results?
  • What public health implication can be expected from the analysis results reported from this study? The discussion on the public health implication from the present findings needs to be strengthened.

Author Response

  • On page 2 line 77, data from 3930 of 9146 invited participants (42.9%) were used for the analysis. The inclusion criteria were adolescents aged 13 to 18 years with complete data on SSB consumption. It would be better to provide detailed sample size information for each age groups (i.e. 6-9 years, 10-12 years and 13-18 years) and percentage of missing data for the variables using in the analysis.

Reply: We have added the requested information to the manuscript. Please see Figure and text in the method section:

The population was divided into 3 age groups (6-9 years, 10,148 students; 10-12 years, 7,035 students; and 13-18 years, 9,146 students) with a specific questionnaire for each age group (Figure 1) (lines:73-75)

  • How was the sampling strategy used in this study? Should the potential intra-class correlations of behavioral outcomes due to students nested within school be considered in the analysis?

Reply: We agree with the reviewer that the cluster effect is an important issue. However, the unit for recruitment was the entire school population enrolled in the educational centers of the City of Sabadell, and therefore, the variability of the population is represented.

  • Rough measurements of physical activity, school performance and screen time were used without justification of measurement methodology and validity.

Reply: The reviewer is correct that physical activity, school performance, and screen time were roughly measured. Due to limited time for data collection it was impossible to administer longer and validated questionnaires. We have modified the limitation section to read:

A limitation of the present study is the use of a non-validated questionnaire of SSB consumption. Furthermore, due to limited time data of physical activity, school performance, and screen time were measured by short not validated questions. This fact could have increased the measurement error inherent in self-reported data”. (lines:227-231)

  • What is the rationale to use failing 3 subjects in the previous school year as an indicator for low school performance?

Reply: To date, there is still no consensus about a definition of school performance and failure. In the present work we have used a normative criterion of the Catalan education system. According to this definition students who do not pass 3 or more assignments should be considered to repeat the course. We have added the corresponding reference:

Reference 12: Diari Oficial de la Generalitat de Catalunya núm 4915, 29-6-2007 Article 19.2. DOGC. Barcelona; 2007. p. 21870-21946 [consultado 5 Dic 2012]. https://dibaaps.diba.cat/vnis/temp/CIDO_dogc_2007_06_20070629_DOGC_20070629_062_138.pdf

  • Other than TV watching, video gaming and computer/tablet use should also be considered for screen time estimate for physical inactivity.  

Reply: The description in the method section is incorrect, We apologize this error. We have modified the description of screen time to read:

Participants also responded to questions on frequency of sport activities, smoking (never, occasionally, regular), sleep duration of at least 8h/day (yes/no), and screen viewing including television, computer, play station, and tablet (never, occasionally, less than 1 hour per day, at least 1 hour but less than 3 hours per day, between 3 hours and less than 5 hours per day, and more than 5 hours per day)” (lines: 92-98)

  • What was the theory or theoretical framework to guide this study and to interpret the analysis results?

Reply: The hypothesis of the present work was that SSB consumption is related to unfavorable lifestyle habits and low socioeconomic status. This hypothesis was confirmed especially in the case of regular soda and energy drink consumption. We have added the hypothesis of our work in the introduction: “The hypothesis of the present work was that SSB consumption is related to unfavorable lifestyle habits and low maternal socioeconomic status.”  (lines:63-64).

  • What public health implication can be expected from the analysis results reported from this study? The discussion on the public health implication from the present findings needs to be strengthened.

Reply: We agree with the reviewer that this is an important issue. We have added the following text to the discussion:  

The results of the present study underline the need for the implementation of health policies with the aim to reduce SSB consumption. Taxation of SSB consumption might be a tool for the reduction of the consumption of these beverages. Currently SSB taxes are implemented in more than 40 countries (26). To date it is too early for a final conclusion of the efficacy of this measure, however, current data are promising (28,29)” (lines: 227-231).

27-      Taxes on sugar-sweetened beverages: International evidence and experience. World Bank Group September 2020. https://openknowledge.worldbank.org/bitstream/handle/10986/33969/Support-for-Sugary-Drinks-Taxes-Taxes-on-Sugar-Sweetened-Beverages-Summary-of-International-Evidence-and-Experiences.pdf?sequence=6.

28-      Sánchez-Romero, L.M.; Canto-Osorio, F.; González-Morales, R.; Colchero, M.A.; Ng, S.W.; Ramírez-Palacios, P.; Salmerón, J.; Barrientos-Gutiérrez, T. Association between Tax on Sugar Sweetened Beverages and Soft Drink Consumption in Adults in Mexico: Open Cohort Longitudinal Analysis of Health Workers Cohort Study. BMJ 2020, 369, doi:10.1136/bmj.m1311.

  1. Bandy, L.K.; Scarborough, P.; Harrington, R.A.; Rayner, M.; Jebb, S.A. Reductions in Sugar Sales from Soft Drinks in the UK from 2015 to 2018. BMC Med. 2020, 18, 20, doi:10.1186/s12916-019-1477-4.

Reviewer 2 Report

Interesting results, would be interested to see how the results compare with neighboring cities in Spain with similar socioeconomic and educational status. I agree, with authors that limitation in the study is the use of a non-validated questionnaire for SSB consumption. Overall good presentation of results and propose publication of this manuscript. 

The total number of students surveyed make this a powerful study. The correlations between higher ED and SSB consumption in lower socioeconomic families, and association with increased smoking and lower physical activity are very important and I would like to see further studies investigating this.   

Suggestions: A universal validated questionnaire needs to be generated for consistency of results and that will be beneficial later when we have more data to conduct a meta-analysis. Especially if the goal is to advocate for policies that will deter the youth from consuming SSB. Neighboring cities in Spain should be included in the survey to serve as a comparison with this study.

Author Response

Interesting results, would be interested to see how the results compare with neighboring cities in Spain with similar socioeconomic and educational status. I agree, with authors that limitation in the study is the use of a non-validated questionnaire for SSB consumption. Overall good presentation of results and propose publication of this manuscript. 

Reply: We thank the reviewer for this assessment of our work.

The total number of students surveyed make this a powerful study. The correlations between higher ED and SSB consumption in lower socioeconomic families, and association with increased smoking and lower physical activity are very important and I would like to see further studies investigating this.   

Reply: Currently we are addressing this issue in the PASOS cohort, a nationwide representative sample of around 4.000 Spanish children and adolescents.

Suggestions: A universal validated questionnaire needs to be generated for consistency of results and that will be beneficial later when we have more data to conduct a meta-analysis. Especially if the goal is to advocate for policies that will deter the youth from consuming SSB. Neighboring cities in Spain should be included in the survey to serve as a comparison with this study.

Reply:  We agree with the reviewer that a universal validated questionnaire is needed for better comparability among studies.

Round 2

Reviewer 1 Report

The authors did an excellent job of addressing the initial comments and questions raised in the first submission. The revised manuscript is well written with clear interpretation of analysis results and adequate highlights of study strengths and limitations.